# Sparse Structure Search for Delta Tuning

**Shengding Hu**[1]*, **Zhen Zhang**[1]*, **Ning Ding**[1], **Yadao Wang**[3],
**Yasheng Wang**[3], **Zhiyuan Liu**[1,2,4]†, **Maosong Sun**[1,2,4]
[1]Dept. of Comp. Sci. & Tech., Institute for AI, Tsinghua University, Beijing, China
Beijing National Research Center for Information Science and Technology
[2]Institute Guo Qiang, Tsinghua University, Beijing, China [3]Noah's Ark Lab, Huawei
[4]International Innovation Center of Tsinghua University, Shanghai, China
{hsd20, zhen-zha19}@mails.tsinghua.edu.cn

## Abstract

Adapting large pre-trained models (PTMs) through fine-tuning imposes prohibitive computational and storage burdens. Recent studies of delta tuning (DT), i.e., parameter-efficient tuning, find that only optimizing a small portion of parameters conditioned on PTMs could yield on-par performance compared to conventional fine-tuning. Generally, DT methods exquisitely design delta modules (DT modules) which could be applied to arbitrary fine-grained positions inside PTMs. However, the effectiveness of these fine-grained positions largely relies on sophisticated manual designation, thereby usually producing sub-optimal results. In contrast to the manual designation, we explore constructing DT modules in an automatic manner. We automatically **S**earch for the **S**parse **S**tructure of **Delta** Tuning ($S^3$Delta). Based on a unified framework of various DT methods, $S^3$Delta conducts the differentiable DT structure search through bi-level optimization and proposes shifted global sigmoid method to explicitly control the number of trainable parameters. Extensive experiments show that $S^3$Delta surpasses manual and random structures with less trainable parameters. The searched structures preserve more than 99% fine-tuning performance with 0.01% trainable parameters. Moreover, the advantage of $S^3$Delta is amplified with extremely low trainable parameters budgets (0.0009%~0.01%). The searched structures are transferable and explainable, providing suggestions and guidance for the future design of DT methods. Our codes are publicly available at `https://github.com/thunlp/S3Delta`.

## 1 Introduction

Increasingly large pre-trained models (PTMs) [6, 27, 30, 31, 12] building upon Transformers [36] have been emerging and achieving state-of-the-art results on a variety of downstream tasks. Despite the blessing of effectiveness, these big models also bring the curse of prohibitive costs on computation and storage during the adaptation because of the gradient computation of the whole model and the giant size of the fine-tuned checkpoint.

To alleviate such costs, studies of delta tuning (DT) [7], also known as parameter-efficient tuning [15, 28, 42, 16, 25, 9, 20, 18], have been developed, which only train a small portion of PTMs and keep the vast majority of parameters frozen. Studies have verified that delta tuning could achieve competitive performance compared to conventional fine-tuning with very few trainable parameters, resulting in considerable savings in model adaptation costs. Generally, these approaches manually design delta modules (DT modules) to complete model adaptation. For example, adapter-based

---

* Equal contribution, ordered alphabetically.
† Corresponding authors: Z.Liu (liuzy@tsinghua.edu.cn)

methods [15, 28, 25] inject two newly-introduced feed-forward layers to Transformers and only fine-tune 0.5%-8% parameters to yield promising results; BitFit [42] only fine-tunes the bias terms (0.04% - 0.1% parameters) within Transformers; LoRA [16] inserts trainable rank decomposition matrices to each layer of Transformers and is successfully adopted on GPT-3 [2] with 175 billion parameters.

While early research focused on how to design practically effective DT modules, more recent research has advanced the understanding of delta tuning more deeply. He et al. [13] bridge connections among different approaches to form a unified framework. And Ding et al. [7] indicate that the combination of different trainable modules could bring different levels of gain on downstream tasks. The above empirical evidence implies that there may exist *an optimal mixture of DT modules* that is more effective than manually designed structures. In fact, considering the fine-grained structure inside PTMs, the positions where the DT modules could be applied are numerous, but not all DT modules at all positions contribute equally to the task performance. How to find *the optimal structure of DT modules* and remove the redundancy in the trainable parameters is essential for a more efficient adaption method. Predictably, such optimal structure is difficult to construct artificially and may vary with specific tasks and models. Therefore, we propose to automatically search the optimal structure that contains a mixture of DT modules at diverse positions inside PTMs. Also importantly, the structure should be sparse to ensure the parameter efficiency.

We present **S**parse **S**tructure **S**earch for **Delta** Tuning ($S^3$Delta) to automatically search such optimal trainable structure, which could flexibly control the number of trainable parameters according to practical requirements. The searching process and the optimization of $S^3$Delta is guided by performance to ensure the effectiveness on specific tasks. Moreover, the structures change automatically to suit the preset limitation of the number of trainable parameters. In contrast, heuristically designed structures are usually coarse-grained and independent of performance and budget, making them neither optimal nor flexible to adjust the number of trainable parameters.

In terms of the specific methodology, we firstly construct a unified search space by applying probabilistic gating controlled by structural parameters to all potential DT modules. Then, we develop a framework of differentiable DT structure search by treating the problem as a constrained neural architecture search problem. In our framework, the structural parameters are updated via bi-level optimization [22]. Unlike the traditional neural architecture search that learns from scratch, we implement the first neural structure search based on a pre-defined backbone and under the delta tuning scenario. To search under a pre-defined budget of trainable parameters, we develop a *shifted global sigmoid* to explicitly control the number of activated DT modules in the searching phase.

We conducted extensive experiments to study the effectiveness of $S^3$Delta. Firstly, the experiments show that with 0.01% parameters, we are able to recover 99% and 98% fine-tuning performance on GLUE [38] and SuperGLUE [37], respectively. Secondly, the searched structure surpasses the human-designed structures considerably while consuming less ($\sim 1/5$) trainable parameters. Moreover, the advantage enlarges when the number of trainable parameters is minimal (0.0009%$\sim$ 0.01%). Furthermore, the searched structures are *transferable* across tasks, which significantly strengthens the usefulness of the searched structures. Apart from the performance boost, we visualize and explain the searched structures, which is beneficial to the future design of new DT methods.

## 2  Related Work

**Delta Tuning (DT).** Our work is related to the studies of delta tuning (DT) for pre-trained models [7]. Generally, DT only optimizes a small portion of parameters and leaves the vast majority of parameters untouched for the adaptation to downstream tasks. Pioneer work select parts of the PTMs to be trainable [35, 11, 10]. Adapter [15] is one of the earliest methods that apply the concept of parameter-wise efficiency to pre-trained language models, which inserts linear neural modules to every Transformer layer and achieves on-par results to full fine-tuning. As the PTMs scaling in recent years, DT is valued for its efficiency in computing and storage. This has spawned not only empirical studies [28, 14] and variants on the adapter [28, 25, 34], but also a range of other approaches. Prefix tuning [20] prepends embeddings to the hidden states of the Transformer model, and prompt tuning [18] further simplifies the strategy and only prepends such embeddings to the input layer. There are also approaches which specify some of the parameters inside PTMs that can be trainable to achieve good results, such as Masking [43], BitFit [42], DiffPruning [9], etc. LoRA [16] assumes that the change in model weights is intrinsically low-rank after fine-tuning, and uses trainable rank-decomposition matrices for model adaptation. In addition to specific methods, some

studies have comprehensively investigated DT methods. He et al. [13] models multiple methods in a unified manner, Ding et al. [7] provides a theoretical discussion and comprehensive empirical study of these methods. Our work proposes to automatically search for trainable structures in the context of DT, which is a different perspective from all the aforementioned work. In terms of the structure of DT, AdapterDrop [33] explores dropping a fraction of Adapter modules based on manual trials. A concurrent work [26] learns switches on Adapter modules to select the beneficial adapter modules. However, it is not optimized under a preset number of trainable parameters. They are also both limited to adapter-based methods. On the contrary, our proposed method can search within a mixture of almost all DT modules under a constrained trainable parameter budget.

**Neural Architecture Search (NAS).** Our work conducts structure search in the scope of DT, which is related to the Neural Architecture Search algorithms. A line of NAS algorithms uses Reinforcement Learning or Evolutionary Algorithms to explore the best structure with reward from training the structure from scratch [44, 45, 32, 29], which usually consumes prohibitive computation resources. Another line of NAS algorithms [22, 21, 5] approaches the problem with gradient-based optimization. DARTS [22] relaxes the discrete structure using continuous structural parameters, which are optimized with gradient-based optimization. DARTS achieves competitive performances with much fewer computational resources. We take inspirations from DARTS in optimizing the structural parameters of $S^3$Delta. We are also the first to conduct NAS conditioned on a pre-trained backbone model. We also take inspiration from the NAS algorithms with binary gates [3, 40].

# 3  Method

In this section, we firstly introduce the preliminaries of pre-trained model adaptation, transformer architecture, and the delta tuning. Then we introduce our method $S^3$Delta in detail.

## 3.1  Preliminaries

**Pretrained Model Adaptation.** The recent prevalent pre-train then fine-tune paradigm in deep learning takes advantage of a pre-trained model $\mathcal{M}$ with parameters $\Theta$ and continues to optimize $\Theta$ on a downstream task $\mathcal{D} = \{\mathcal{D}_{\text{train}}, \mathcal{D}_{\text{val}}, \mathcal{D}_{\text{test}}\}$ under an objective function $\mathcal{L}$. In fine-tuning, all the parameters of the pre-trained model are optimized using the train split to minimize $\mathcal{L}$, i.e.,

$$\min_{\Theta} \mathcal{L}(M(\Theta), \mathcal{D}_{\text{train}}). \tag{1}$$

**Transformer Architecture.** The pre-trained models typically adopt the Transformer model [36] as their backbone. The Transformer model is composed of multiple stacked Transformer layers that processes the hidden state sequentially through different computation modules, such as Self-Attention module (SelfAttn), Cross-Attention module (CrossAttn), Feed-Forward module (FFN), and Layer Normalization module (LN), etc., and details of each module are in Appendix A. The computation process in the Transformer can be abstracted by a sequence of transformations of the hidden representation. In each computation step, the input hidden representation $\mathbf{H}^{\text{in}} \in \mathbb{R}^{s \times d_1}$ is transformed into an output hidden representation $\mathbf{H}^{\text{out}} \in \mathbb{R}^{s \times d_2}$, where $s$ is the sequence length of the input and $d_1, d_2$ are the hidden dimensions,

$$\mathbf{H}^{\text{out}} = m(\mathbf{H}^{\text{in}}). \tag{2}$$

**Delta Tuning (DT).** DT methods only train a small portion of parameters conditioned on the backbone PTMs to improve the adaptation efficiency [15, 28, 42, 16, 25, 9, 20, 18]. Although the specific forms of the various DT modules are substantially different, He et al. [13] unify them as modifications $\Delta$ of the hidden state [3],

$$\mathbf{H}^{\text{out}} = m(\mathbf{H}^{\text{in}}) + \Delta. \tag{3}$$

The formulas of some DT methods under the unified view are listed in Table 1. The DT modules can be applied to extensive positions on the backbone PTMs, which are listed in the rightmost column in Table 1. In training, we freeze all the parameters in the backbone module $m$, i.e., $\Theta$, and set

---

[3]We use a little bit more flexible notation than [13], which takes into account the frozen backbone module $m$ and thus can distinguish the DT modules that take either $\mathbf{H}^{\text{out}}$ or $\mathbf{H}^{\text{in}}$ as their input.

Table 1: Different DT methods are the specializations of the unified view (Equation (3)) and can be applied to extensive positions on the PTMs.

| Method | Transformation | $\Delta$ | Potential Positions |
|---|---|---|---|
| LoRA [16] | $\mathbf{H}^{\text{out}} = \mathbf{H}^{\text{in}}(\mathbf{W} + \mathbf{AB})$ | $\mathbf{H}_0\mathbf{AB}$ | Weight matrices |
| Adapter [15] | $\mathbf{H}^{\text{out}} = m(\mathbf{H}^{\text{in}}) + f(m(\mathbf{H}^{\text{in}})\mathbf{W}_{\text{down}})\mathbf{W}_{\text{up}}$ | $f(m(\mathbf{H}^{\text{in}})\mathbf{W}_{\text{down}})\mathbf{W}_{\text{up}}$ | After any modules |
| Parallel Adapter [13] | $\mathbf{H}^{\text{out}} = m(\mathbf{H}^{\text{in}}) + f(\mathbf{H}^{\text{in}}\mathbf{W}_{\text{down}})\mathbf{W}_{\text{up}}$ | $f(\mathbf{H}^{\text{in}}\mathbf{W}_{\text{down}})\mathbf{W}_{\text{up}}$ | Between any two modules |
| BitFit [42] | $\mathbf{H}^{\text{out}} = m(\mathbf{H}^{\text{in}}) + \mathbf{b}_\delta$ | $\mathbf{b}_\delta$ | Linear layers |
| LNFit [4] | $\mathbf{H}^{\text{out}} = \frac{\mathbf{H}^{\text{in}}}{\text{Var}(\mathbf{H}^{\text{in}})}(\mathbf{s} + \mathbf{s}_\delta) + \mathbf{b}$ | $\frac{\mathbf{H}^{\text{in}}}{\text{Var}(\mathbf{H}^{\text{in}})}\mathbf{s}_\delta$ | Layer Normalization modules |

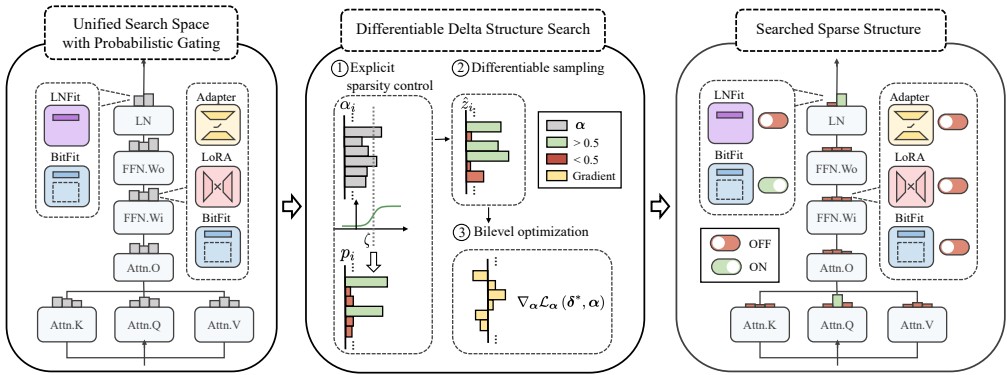

Figure 1: The framework of S³Delta. We propose a unified search space with probabilistic gating to enable search among a mixture of DT methods. We find the optimal sparse structure using the differentiable DT structure search and explicit sparsity control.

parameters introduced in computing $\Delta$, denoted by $\boldsymbol{\delta}$, as the only trainable parameters. Therefore, the adaptation objective in DT is

$$\min_{\boldsymbol{\delta}} \mathcal{L}(\mathcal{M}(\Theta, \boldsymbol{\delta}), \mathcal{D}_{\text{train}}). \tag{4}$$

For the convenience of notation, we simplify Equation (4) into

$$\min_{\boldsymbol{\delta}} \mathcal{L}_{\text{train}}(\boldsymbol{\delta}). \tag{5}$$

### 3.2 Sparse Structure Search for Delta Tuning

Our target is to search for the optimal structure of DT constrained by a pre-defined and limited trainable parameter budget $\mathcal{B}$. To achieve this, we design Sparse Structure Search for Delta Tuning (S³Delta), which is driven by three essential components: a *unified search space with probabilistic gating*, an efficient *differentiable DT structure search* algorithm, and an *explicit sparsity control algorithm using shifted global sigmoid*.

**Unified Search Space with Probabilistic Gating**. The potential positions in the backbone models to which DT modules could be applied are extensive, especially when we consider a mixture of different types of DT modules through the unified view (Equation (3)). However, not all positions contribute to the task performance equally, and only a fraction of positions should be *activated* to avoid redundancy of the trainable parameters. To this end, we design a *probabilistic gating* mechanism over all possible DT modules' positions. Specifically, for each DT module that computes $\Delta_i$ for the hidden representation, we *activate* the modification $\Delta_i$ with probability $p_i \in [0, 1]$,

$$\mathbf{H}_i^{\text{out}} = m(\mathbf{H}_i^{\text{in}}) + z_i\Delta_i, \tag{6}$$

where $z_i \in \{0, 1\} \sim B(1, p_i)$ is a random sample from Bernoulli distribution.

**Differentiable DT Structure Search.** Finding the optimal structure from a search space with abundant potential positions is challenging due to the compositionality of activated positions. Furthermore,

---

[4]LNFit only trains the variance vector in the Layer Normalization module of the PTMs, which is inspired by Frankle et al. [8] who only train the Batch Normalization module in Convolutional Neural Networks.

directly comparing the fully trained model with each DT structure is intractable. In our work, we propose to *optimize the gating probability $p_i$ with gradient-based optimization*. To make the sampling process differentiable, we use the Binary Concrete Distribution [24, 17] as a soft and differentiable approximation to the Bernoulli distribution,

$$\hat{z}_i = \sigma\left(\frac{1}{\beta}\log\frac{up_i}{(1-u)(1-p_i)}\right), \tag{7}$$

where $u \sim U(0,1)$ is a random sample from the uniform distribution in $[0,1]$ and $\beta$ is the temperature to control the sharpness of the distribution $\hat{z}_i$ distribution [5]. Similar distributions are used in learning sparse networks [23] or pruning dense networks [39]. By replacing the hard sample $z_i$ with the soft approximation $\hat{z}_i$, we can back-propagate through $p_i$ in training. However, directly optimizing $p_i$ in the probability space $[0,1]$ may lead to numerical instability, therefore, we parameterize it with structural parameters $\alpha_i \in \mathbb{R}$, i.e., $p_i = g(\alpha_i)$. And we denote all the structure parameters as $\boldsymbol{\alpha}$.

We optimize $\boldsymbol{\alpha}$ through bi-level optimization [1, 22], i.e., optimizing $\boldsymbol{\alpha}$ conditioned on the optimized parameters $\boldsymbol{\delta}^*$ of the DT modules. The inner and outer level of optimization are conducted on separate splits of training data, denoted by $\mathcal{D}_{\boldsymbol{\delta}}, \mathcal{D}_{\boldsymbol{\alpha}}$, which is analogous to validating structures trained on $\mathcal{D}_{\boldsymbol{\delta}}$ using a different split $\mathcal{D}_{\boldsymbol{\alpha}}$ to avoid over-fitting to $\mathcal{D}_{\boldsymbol{\delta}}$. Thus, the optimization objective is

$$\min_{\boldsymbol{\alpha}} \mathcal{L}_{\boldsymbol{\alpha}}(\mathcal{M}(\Theta, \boldsymbol{\delta}^*, \boldsymbol{\alpha})), \tag{8}$$

$$s.t. \ \boldsymbol{\delta}^* = \operatorname{argmin}_{\boldsymbol{\delta}} \mathcal{L}_{\boldsymbol{\delta}}(\mathcal{M}(\Theta, \boldsymbol{\delta}, \boldsymbol{\alpha})). \tag{9}$$

Following DARTS [22], we make approximations to the gradient of the structural parameters by applying chain rule and taking finite difference approximations [6],

$$\nabla_{\boldsymbol{\alpha}} \mathcal{L}_{\boldsymbol{\alpha}}\left(\boldsymbol{\delta}^*, \boldsymbol{\alpha}\right) \tag{10}$$

$$\approx \nabla_{\boldsymbol{\alpha}} \mathcal{L}_{\boldsymbol{\alpha}}\left(\boldsymbol{\delta} - \xi\nabla_{\boldsymbol{\delta}}\mathcal{L}_{\boldsymbol{\delta}}(\boldsymbol{\delta},\boldsymbol{\alpha}), \boldsymbol{\alpha}\right) \tag{11}$$

$$\approx \nabla_{\boldsymbol{\alpha}} \mathcal{L}_{\boldsymbol{\alpha}}\left(\boldsymbol{\delta}', \boldsymbol{\alpha}\right) - \xi\nabla^2_{\boldsymbol{\alpha},\boldsymbol{\delta}}\mathcal{L}_{\boldsymbol{\delta}}(\boldsymbol{\delta},\boldsymbol{\alpha})\nabla_{\boldsymbol{\delta}'}\mathcal{L}_{\boldsymbol{\alpha}}\left(\boldsymbol{\delta}', \boldsymbol{\alpha}\right) \tag{12}$$

$$\approx \nabla_{\boldsymbol{\alpha}} \mathcal{L}_{\boldsymbol{\alpha}}(\boldsymbol{\delta}', \boldsymbol{\alpha}) - \xi\frac{\nabla_{\boldsymbol{\alpha}}\mathcal{L}_{\boldsymbol{\delta}}(\boldsymbol{\delta}^+, \boldsymbol{\alpha}) - \nabla_{\boldsymbol{\alpha}}\mathcal{L}_{\boldsymbol{\delta}}(\boldsymbol{\delta}^-, \boldsymbol{\alpha})}{2\epsilon}, \tag{13}$$

where the optimal $\boldsymbol{\delta}^*$ is approximated by the parameters of one-step update $\boldsymbol{\delta}' = \boldsymbol{\delta} - \xi\nabla_{\boldsymbol{\delta}}\mathcal{L}_{\boldsymbol{\delta}}(\boldsymbol{\delta}, \boldsymbol{\alpha})$. $\xi$ is the learning rate of parameters $\boldsymbol{\delta}$, and $\epsilon$ is a small scalar used in the finite difference approximation.

**Explicit Sparsity Control with Shifted Global Sigmoid.** Most of the DT modules in the search space are redundant and contribute little to the performance. However, the search algorithm may not be aware of the sparsity target and degenerate to greedily adding more DT modules. As opposed to previous sparse network learning methods [23, 9] which punish the dense structures with $L_0$ regularization, we explicitly control the sparsity of structure at the target level during the search through a *shifted global sigmoid* parameterization. (See Section 4.9 for comparing the two methods),

$$p_i = \tilde{p}_i \frac{\sum_i \operatorname{Detach}(\tilde{p}_i)}{\sum_i \tilde{p}_i}, \tag{14}$$

$$\text{where} \quad \tilde{p}_i = \operatorname{Sigmoid}(\frac{\alpha_i - \zeta}{\tau}). \tag{15}$$

The $\operatorname{Detach}(\cdot)$ operator turns a parameter that requires gradient into a scalar that is free from gradient computation. Equation (14) doesn't change the value of $\tilde{p}_i$, but it enforces the competition among different positions and DT modules, which is similar to $\operatorname{Softmax}$ operation (See Appendix C.2 for details).

---

[5]The distribution of $\hat{z}_i$ has the property that $P(\hat{z}_i > 0.5) = p_i$, and when $\beta$ approximate 0, the distribution of $\hat{z}_i$ converges to $B(1, p_i)$ (See Appendix C.1), which makes it a suitable surrogate for Bernoulli distribution.

[6]We use the same $\hat{z}_i$ sample to compute $\nabla_{\boldsymbol{\alpha}}\mathcal{L}_{\boldsymbol{\delta}}(\boldsymbol{\delta}^+, \boldsymbol{\alpha})$, $\nabla_{\boldsymbol{\alpha}}\mathcal{L}_{\boldsymbol{\delta}}(\boldsymbol{\delta}^-, \boldsymbol{\alpha})$, $\nabla_{\boldsymbol{\delta}}\mathcal{L}_{\boldsymbol{\delta}}(\boldsymbol{\delta}, \boldsymbol{\alpha})$, and $\nabla_{\boldsymbol{\alpha}}\mathcal{L}_{\boldsymbol{\alpha}}\left(\boldsymbol{\delta}', \boldsymbol{\alpha}\right)$.

In Equation (15), $\zeta$ is a scalar. Increasing $\zeta$'s value will monotonically reduce $p_i$ to 0 while keeping $p_i$ in $[0, 1]$. So the expected number of trainable parameters $\mathbb{E}[N]$ is a monotonic function w.r.t. $\zeta$,

$$\mathbb{E}[N] = \mathbb{E}\left[\sum_i \mathbb{I}(z_i = 1)|\boldsymbol{\delta}_i|\right] \approx \mathbb{E}\left[\sum_i \mathbb{I}(\hat{z}_i > 0.5)|\boldsymbol{\delta}_i|\right] = \sum_i p_i|\boldsymbol{\delta}_i|, \quad (16)$$

where the $|\boldsymbol{\delta}_i|$ is the number of parameters introduced in computing $\Delta_i$. Thus, we can dynamically adjust $\zeta$ to make $\mathbb{E}[N]$ approach $\mathcal{B}$ via monotonic optimization,

$$\zeta^* = \operatorname{argmin}_\zeta(\sum_i p_i|\boldsymbol{\delta}_i| - \mathcal{B}), \quad \text{where} \quad \sum_i p_i|\boldsymbol{\delta}_i| \leq \mathcal{B}. \quad (17)$$

**Evaluation of the Searched Structure.** To determine the final structure of DT, instead of sampling from $p_i$, we choose the set of positions where the sum of $p_i$ is the highest while still being within the budget $\mathcal{B}$. This deterministic algorithm reduces the variance of the final structures. After obtaining the final structure, we re-initialize and re-train the parameters in the DT modules to converge on $\mathcal{D}_{\text{train}}$.

---

**Algorithm 1** Algorithm of S$^3$Delta

---

Initialize all DT modules in the search space, and initialize $\boldsymbol{\alpha}$.
**while** *not converged* **do**
    1. Calculate $\zeta$, $p_i$, and sample $\hat{z}_i$.
    2. Compute the each loss terms by forward and backward propagation.
    3. Update $\boldsymbol{\alpha}$ according to Equation (13).
    4. Update $\boldsymbol{\delta}$ using $\nabla_{\boldsymbol{\delta}}\mathcal{L}_{\boldsymbol{\delta}}(\boldsymbol{\delta}, \boldsymbol{\alpha})$.
**end while**
Determine and evaluate the final structure.

---

# 4 Experiments

## 4.1 Datasets and PTMs

We apply S$^3$Delta to multitask benchmarks GLUE [38] and SuperGLUE [37] following previous works. All datasets are downloaded from the HuggingFace Datasets [19]. Since the test splits of these datasets are held officially and invisible to the researchers, we conduct random splits from either train set or validation set to make the new train, validation, and test splits, which is critical to ensure fair evaluations according to Chen et al. [4]. We repeat 4 times using different random seeds for experiments in Table 2, and 8 times for experiments in Figure 2. The details are in Appendix B. We use the T5$_{\text{large}}$ model (703M parameters) as the backbone PTMs.

## 4.2 Baselines

We compare S$^3$Delta with several widely used baselines (See Appendix B for details).

**Fine-tune.** Traditional fine-tuning trains all parameters in the PTMs.

**LoRA.** We apply LoRA linear layer to the Self-Attention's query modules and value modules as Hu et al. [16] suggest. We include two rank levels ($r = 8$ and $r = 1$) in our experiments.

**Adapter.** We adopt the first adapter method proposed by Houlsby et al. [15]. Their method requires more parameters than the other methods but achieves good empirical results.

**Low Rank Adapter (Adapter-LR).** We adopt the Low Rank Adapter as an efficient variant of the adapter-based method. It is proposed in [25] as a simple but effective baseline. The rank is set to 1.

**BitFit.** BitFit proposes to only adapt the bias layer in the model. We adopt the same setting as Zaken et al. [42] that tunes the bias inside all linear modules, and the Layer Normalization layer[7].

**LNFit.** We train the variance vector of all Layer Normalization layers', including the Layer Normalization after the whole transformer encoder.

---

[7]Although T5 has no bias in linear modules, we can treat it as bias vectors with zero initialization.

Table 2: Results on GLUE [38] benchmark (above) and SuperGLUE [37] benchmark (below). Green and blue represent the best and second best scores, respectively, among the methods in our search space. The first three rows represent the results of fine-tuning and other DT methods, which are not used in our search space due to high trainable parameter ratios. On SuperGLUE tasks, since the results on COPA vary dramatically ($\pm 26.00$), the average results of SuperGLUE become easily dominated by the results on COPA. Therefore we also report the average results that exclude COPA ($\text{AVG}_{-\text{COPA}}$). The widths of the yellow rectangles are proportional to the trainable parameter ratios.

| GLUE | | | | | | | | | |
|---|---|---|---|---|---|---|---|---|---|
| Parameter Ratios | Method | CoLA | SST2 | MRPC | QQP | STSB | MNLI | QNLI | AVG |
| 10000%% | Fine-tune | 62.25 ± 3.96 | 95.87 ± 0.42 | 91.86 ± 1.19 | 89.50 ± 0.22 | 91.86 ± 0.46 | 89.61 ± 0.30 | 94.22 ± 0.35 | 87.88 |
| 65.33%% | Adapter | 59.03 ± 3.06 | 95.90 ± 0.29 | 93.02 ± 0.28 | 88.39 ± 0.06 | 91.77 ± 0.25 | 89.53 ± 0.07 | 94.17 ± 0.19 | 87.40 |
| 21.32%% | LoRA($r$=8) | 58.43 ± 4.16 | 95.79 ± 0.27 | 92.21 ± 0.88 | 88.35 ± 0.25 | 91.78 ± 0.31 | 89.38 ± 0.32 | 94.14 ± 0.12 | 87.15 |
| | | Methods in the Search Space | | | | | | | |
| 8.13%% | BitFit | 56.98 ± 3.89 | 96.24 ± 0.33 | 92.16 ± 0.68 | 88.12 ± 0.07 | 91.59 ± 0.08 | 89.10 ± 0.09 | 94.07 ± 0.21 | 86.90 |
| 4.12%% | Adapter-LR | 56.78 ± 4.80 | 95.90 ± 0.14 | 92.76 ± 0.67 | 88.08 ± 0.13 | 91.26 ± 0.31 | 89.30 ± 0.14 | 93.94 ± 0.07 | 86.86 |
| 2.67%% | LoRA($r$=1) | 56.77 ± 2.29 | 95.81 ± 0.27 | 92.45 ± 1.00 | 88.08 ± 0.11 | 91.54 ± 0.33 | 89.16 ± 0.17 | 94.10 ± 0.05 | 86.84 |
| 1.70%% | LNFit | 56.15 ± 4.06 | 95.81 ± 0.20 | 91.71 ± 0.39 | 88.17 ± 0.10 | 91.37 ± 0.24 | 89.11 ± 0.09 | 93.99 ± 0.20 | 86.62 |
| 1.39%% | S³Delta-M | 59.34 ± 4.75 | 95.84 ± 0.14 | 92.13 ± 2.09 | 88.04 ± 0.23 | 91.58 ± 0.25 | 89.14 ± 0.13 | 94.12 ± 0.12 | 87.17 |
| 1.39%% | S³Delta-L | 56.71 ± 3.03 | 95.93 ± 0.15 | 93.27 ± 1.39 | 88.14 ± 0.08 | 91.58 ± 0.49 | 88.81 ± 0.44 | 93.95 ± 0.11 | 86.91 |
| 0.35%% | S³Delta-M | 54.56 ± 3.66 | 95.93 ± 0.24 | 92.14 ± 1.10 | 88.02 ± 0.20 | 91.38 ± 0.34 | 89.04 ± 0.25 | 93.93 ± 0.14 | 86.43 |

| SuperGLUE | | | | | | | | | |
|---|---|---|---|---|---|---|---|---|---|---|
| Parameter Ratios | Method | BoolQ | CB | COPA | MultiRC | ReCORD | RTE | WIC | AVG | AVG$_{-\text{COPA}}$ |
| 10000%% | Fine-tune | 86.67 ± 0.21 | 96.43 ± 2.92 | 73.50 ± 5.26 | 76.65 ± 1.01 | 85.03 ± 0.67 | 88.49 ± 2.12 | 73.12 ± 1.71 | 82.84 | 84.40 |
| 65.33%% | Adapter | 85.98 ± 0.68 | 94.64 ± 6.19 | 63.00 ± 7.75 | 77.60 ± 0.84 | 85.96 ± 0.37 | 89.21 ± 2.94 | 71.63 ± 0.90 | 81.15 | 84.17 |
| 21.32%% | LoRA($r$=8) | 85.06 ± 0.70 | 91.96 ± 3.42 | 51.00 ± 4.16 | 76.94 ± 1.16 | 85.84 ± 0.21 | 87.05 ± 0.59 | 72.10 ± 1.31 | 78.56 | 83.16 |
| | | Methods in the Search Space | | | | | | | | |
| 8.13%% | BitFit | 85.02 ± 0.48 | 89.29 ± 2.92 | 75.00 ± 8.08 | 75.79 ± 1.15 | 85.85 ± 0.32 | 86.15 ± 1.48 | 72.34 ± 1.61 | 81.35 | 82.41 |
| 4.12%% | Adapter-LR | 84.53 ± 0.37 | 84.82 ± 8.44 | 49.50 ± 6.81 | 76.67 ± 1.37 | 86.04 ± 0.09 | 85.61 ± 2.42 | 71.39 ± 0.70 | 76.94 | 81.51 |
| 2.67%% | LoRA($r$=1) | 85.60 ± 0.45 | 84.82 ± 1.79 | 67.50 ± 5.00 | 76.71 ± 1.05 | 85.95 ± 0.36 | 86.87 ± 1.08 | 71.32 ± 1.29 | 79.82 | 81.88 |
| 1.70%% | LNFit | 84.07 ± 0.50 | 82.14 ± 2.92 | 49.00 ± 1.15 | 75.52 ± 1.16 | 86.14 ± 0.11 | 86.69 ± 1.81 | 69.28 ± 1.49 | 76.12 | 80.64 |
| 1.39%% | S³Delta-M | 84.92 ± 0.68 | 92.86 ± 2.92 | 70.50 ± 3.79 | 76.38 ± 0.92 | 86.10 ± 0.11 | 86.69 ± 1.90 | 71.63 ± 1.07 | 81.30 | 83.10 |
| 1.39%% | S³Delta-L | 85.00 ± 0.67 | 90.18 ± 6.10 | 60.00 ± 9.52 | 76.17 ± 1.41 | 86.02 ± 0.15 | 85.79 ± 1.36 | 71.63 ± 1.39 | 79.26 | 82.46 |
| 0.35%% | S³Delta-M | 83.56 ± 0.53 | 87.50 ± 4.61 | 54.00 ± 4.32 | 76.09 ± 0.97 | 86.10 ± 0.26 | 85.79 ± 1.89 | 68.42 ± 1.89 | 77.35 | 81.24 |

We do not include Prompt Tuning [18] as our baseline because it takes much longer steps to converge and doesn't achieve competitive performance on T5$_{\text{large}}$ [18].

### 4.3 S³Delta Search Spaces and Budgets

The search space has a noteworthy influence on the performance of S³Delta. In our experiment, we define two kinds of search space.

**Mix**. The first search space considers a mixture of LoRA, Adapter-LR, Bitfit, and LNFIT modules. LoRA can be applied to any linear modules in the transformer block, including the query(Q), key(K), value(V), output(O) sub-modules of the attention module(ATTN), and the two sub layer W1 and W2 in Feed Forward modules(FFN). The Adapter-LR can theoretically be applied to any position in the computational graph. However, to avoid overcomplicating the search space, we limit it to the outputs of the ATTN module and the FFN modules. For BitFit, the potential applied positions are all the linear modules (Q, K, V, O, W1, W2) and the Layer Normalization modules (LN). For LNFit, the potential positions are all the LN modules. In this search space, there are 916 potential DT modules to be selected and the total number of candidate structures is $2^{916}$ if we do not consider the budget constraint. We denote the structures searched on this search space as **S³Delta-M**.

**LoRA**. We narrow down the search space into a single type of DT module. We choose LoRA as an example. The potential positions are the same as the LoRA modules in Mix search space. There are 288 potential positions in total. We denote the structures searched on this search space as **S³Delta-L**.

We also explore different numbers of trainable parameters. Experiments in Table 2 are conducted on 1.39%% and 0.35%% trainable parameters ratios. More sparsity levels are tested in section 4.5.

### 4.4 Results on GLUE and SuperGLUE

Table 2 shows the performance of different methods on GLUE and SuperGLUE tasks. Comparing S³Delta with the manual structures within the search space, we find that S³Delta-M (1.39%%) achieves the highest average score on GLUE and SuperGLUE (without COPA) despite using the least number of trainable parameters ($\sim$ 1/5 compared to BitFit). S³Delta-M (0.35%%) also

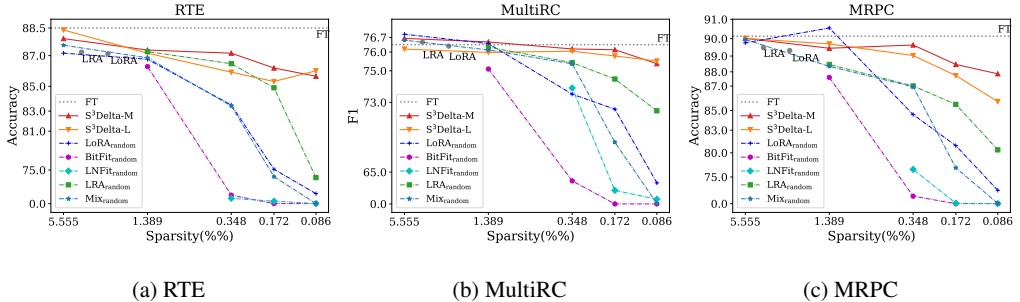

(a) RTE          (b) MultiRC          (c) MRPC

Figure 2: Performances under different trainable parameters ratios. The x-axis represents the ratio of the number of trainable parameters to the backbone PTM's parameters. The scaled y-axis represents the scores. The accuracy of fine-tuning is in gray horizontal line. The result of LoRA ($r$=1) and Low Rank Adapter are plotted in grey dot.

surpasses Adapter-LR, LoRA ($r$=1), LNFit using approximately $1/12, 1/8, 1/5$ trainable parameters, respectively. Narrowing the search space from Mix to LoRA leads to a moderate decrease in performance, which justifies the need to search among a mixture of DT modules. It may also hint that the combination of different DT modules could lead to stronger performance. However, even though the performance of S$^3$Delta-L is not optimal, it compares favorably to LoRA($r$=1), which also shows that the human designed structures, though benefit from applying DT modules uniformly on the PTMs, is sub-optimal. Compared with fine-tuning, S$^3$Delta-M (1.39%%) preserves 99.2% and 98.1% performance on GLUE and SuperGLUE, respectively. In fact, we must emphasize that S$^3$Delta is orthogonal to specific DT modules. The performance of S$^3$Delta can benefit from the future invention of better DT modules, thus potentially achieving comparable or even superior performance to fine-tuning with extremely limited trainable parameters.

## 4.5 Performance under Different Sparsity Levels

To explore the limit of trainable parameter reduction, we train different methods with decreasing sparsity levels from 5.6%% to 0.086%%. To apply baseline methods on target numbers of trainable parameters, we randomly sample a set of potential positions in their corresponding search space to reach the target sparsity level. In Figure 2, we demonstrate the results on three datasets, RTE, MultiRC and MRPC. We can see that S$^3$Delta-M and S$^3$Delta-L have considerable advantages in extremely low trainable parameter budgets. For example, S$^3$Delta-M trains only 0.086%% parameters whereas recovering 96.8%,98.7%,97.5% of the FT performances on RTE, MultiRC, MRPC, respectively. With 5.6%% trainable parameters on MultiRC and MPRC, all the methods saturate to FT performance, proving the feasibility of removing redundant parameters with S$^3$Delta.

## 4.6 Transferability of the Searched Structures

Another essential characteristic of S$^3$Delta is the transferability of the searched structure. In Table 4, we split the GLUE benchmark into source datasets and target datasets. We search on the source dataset (Mix search space) and train the searched structure on the target datasets. We can see that the searched structures are highly transferable, even surpassing the structures direct searched on the target datasets sometimes. The transferability guarantees the reusability of the searched structures.

## 4.7 Efficiency of the Search Process

Although S$^3$Delta focuses on the parameter-efficiency of the searched structures, we also analyze the searching efficiency in Table 3. Generally speaking, the search for an optimal structure consumes 5∼8 times training time and 2 times GPU memory (Due to bi-level optimization). However, it is affordable compared to manually designing different structures and running numerous evaluations.

## 4.8 Visualization and Explanations of the Search Structures

To understand the searched structures, we draw the heat maps of the $p_i$ on different datasets in Appendix D.2. We find obvious patterns and similarities in most datasets. Therefore, we average the $p_i$ across datasets to see the overall pattern of the searched structure. Figure 3 shows the heatmap of $p_i$ of

Table 3: The computational resources in the searching phase and re-training phase, Computation time, memory consumption are listed.

| Dataset | Search | Re-train | Ratio |
|---------|--------|----------|-------|
| Time/min | | | |
| RTE | 148.6 | 30.0 | 5.0 |
| STSB | 139.3 | 26.3 | 5.3 |
| CoLA | 145.6 | 17.0 | 8.6 |
| Memory/GB | | | |
| RTE | 27.7 | 16.6 | 1.7 |
| STSB | 28.9 | 10.6 | 2.7 |
| CoLA | 16.1 | 8.9 | 1.8 |

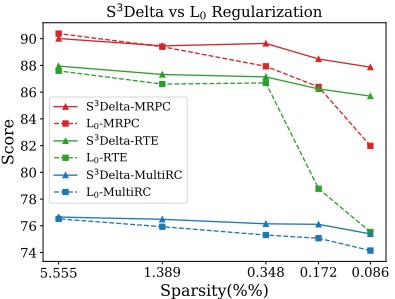

Figure 4: Comparing *shifted global sigmoid* to $L_0$ regularization. Performance on different datasets is in different colors. *Shifted global sigmoid* and $L_0$ *regularization* are in solid lines and dotted lines, respectively.

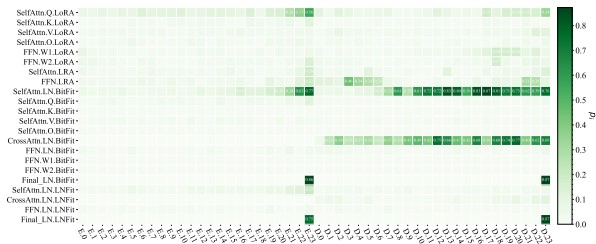

Figure 3: Visualization of $p_i$ of S³Delta-M. The numbers on the squares are the average of $p_i$ across all datasets and all seeds. The deeper the color is, the more activated is the DT module. The x-axis represents different layers of PTMs (E denotes Encoder, and D denotes Decoder), and the y-axis represents different positions (modules) of PTMs.

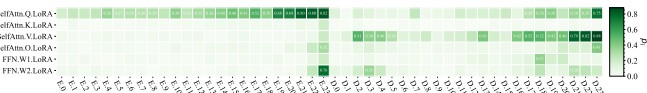

Figure 5: Visualization of $p_i$ of DT modules in S³Delta-L.

Table 4: Structure transfer from source datasets and target datasets. The target datasets are in the row name, and the source datasets are in the column names. "No transfer" means the structure is searched on the target dataset.

| Source | Target Datasets | | | |
|--------|------|-----|------|------|
| | STSB | QQP | QNLI | CoLA |
| No transfer | 91.58 ± 0.25 | 88.03 ± 0.23 | 94.11 ± 0.12 | 59.34 ± 4.75 |
| MRPC | 91.63 ± 0.31 | 88.16 ± 0.08 | 93.96 ± 0.06 | 56.41 ± 3.81 |
| MNLI | 91.39 ± 0.67 | 88.06 ± 0.08 | 94.13 ± 0.10 | 56.38 ± 3.98 |
| SST2 | 91.37 ± 0.23 | 88.02 ± 0.10 | 94.14 ± 0.16 | 55.58 ± 4.13 |

S³Delta-M. We can see that (1) The BitFit modules in the Self-Attention modules and Cross-Attention modules in the higher decoder layers are highly preferred, proving that the BitFit modules are simple and effective. This observation is also beyond the intuition of human experts, as most previous work ignores the contribution of training or applying DT methods to Cross-Attention modules; (2) The last layers of the encoder and decoder are emphasized, which is close to the traditional use of PTMs as feature extractors by training only the last layer; (3) We also observe that BitFit modules tend to be distributed approximately evenly across the higher layers (See Appendix D.2 for details). Figure 5 shows the $p_i$ of S³Delta-L. The trend of choosing higher layers still exists. Interestingly, the query sub-modules are prioritized in the encoder, while the value sub-modules are stressed in the decoder.

### 4.9 Ablation Study

To explicitly control sparsity, we propose *shifted global sigmoid*, which differs from the $L_0$ regularization used in previous work [23]. We compare the results of regularization using *shifted global sigmoid* and $L_0$ on three datasets. From Figure 4, it is clear that *shifted global sigmoid* has an advantage over $L_0$ regularization at almost all sparsity, and the advantage increases with increasing sparsity.

## 5 Conclusion

In this paper, we propose Sparse Structure Search for Delta Tuning (S³Delta), which conducts differentiable DT structure search with explicit sparsity control in a unified search space of a mixture of various DT modules. Experiments demonstrate the effectiveness of S³Delta to find the optimal structure of DT modules and push the limit of trainable parameter reduction. For future works, there are open questions that are worth investigating. (1) Better search spaces or better DT modules could be designed to further explore the potential of structure search. (2) The current NAS algorithms are not tailored for the scenario where a pre-trained backbone model exists. Therefore, more specialized search algorithms could be developed for DT structure search.

# 6 Acknowledgements

This work is supported by National Key R&D Program of China (No. 2020AAA0106502), Institute Guo Qiang at Tsinghua University, Beijing Academy of Artificial Intelligence (BAAI), International Innovation Center of Tsinghua University, Shanghai, China.

Shengding Hu proposed the idea and framework. Shengding Hu and Zhen Zhang designed the methods and experiments. Zhen Zhang conducted the experiments. Shengding Hu and Ning Ding wrote the paper. Zhiyuan Liu and Maosong Sun advised the project and participated in the discussion. Yadao Wang and Yasheng Wang participated in the discussion and provided computational resources.

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
