# A Transformer Architecture

## A.1 Transformer Architecture

The pre-trained language model typically adopts a Transformer model [36] as their backbone, which is formed by stacking multiple transformer layers. The parameters of a transformer layer are located in three sub-modules. The self-attention sub-layer captures the interactions between the $s$ input tokens' hidden representations $\mathbf{H}_{\text{in}}^{(l)} \in \mathbb{R}^{s \times d}$

$$\mathbf{H}_{\text{attn}}^{(l)} = \text{Softmax}(\frac{\mathbf{Q}\mathbf{K}^{\top}}{\sqrt{d}})\mathbf{V}\mathbf{W}_o \tag{18}$$

where

$$\mathbf{Q} = \mathbf{H}_{\text{in}}^{(l)}\mathbf{W}_q^{(l)}; \mathbf{K} = \mathbf{H}_{\text{in}}^{(l)}\mathbf{W}_k^{(l)}; \mathbf{V} = \mathbf{H}_{\text{in}}^{(l)}\mathbf{W}_v^{(l)}; \mathbf{W}_q^{(l)}, \mathbf{W}_k^{(l)}, \mathbf{W}_v^{(l)}, \mathbf{W}_o^{(l)} \in \mathbb{R}^{d \times d} \tag{19}$$

Then the output is followed by the fully connected feed-forward layer, which is typically composed of two linear projections

$$\mathbf{H}_{\text{ff}}^{(l)} = \sigma(\mathbf{H}_{\text{attn}}^{(l)}\mathbf{W}_1^{(l)} + \mathbf{b}_1^{(l)})\mathbf{W}_2^{(l)} + \mathbf{b}_2^{(l)} \tag{20}$$

where $\mathbf{W}_1 \in \mathbb{R}^{d \times d_m}, \mathbf{b_1} \in \mathbb{R}^{d_m}, \mathbf{W}_2 \in \mathbb{R}^{d_m \times d}, \mathbf{b}_2 \in \mathbb{R}^d$, and generally $d_m \geq d$. $\sigma(\cdot)$ is an activation function. Suppose we use a post-norm structure [31, 41], the final sublayer is a layernorm sublayer to add normalize the hidden states

$$\mathbf{H}_{\text{out}}^{(l)} = \text{LayerNorm}(\mathbf{H}_{\text{ff}}^{(l)}) = \frac{\mathbf{H}_{\text{ff}}^{(l)}}{\text{var}(\mathbf{H}_{\text{ff}})}\mathbf{s}^{(l)} + \mathbf{b}^{(l)} \tag{21}$$

where $\mathbf{s}^{(l)}, \mathbf{b}^{(l)} \in \mathbb{R}^d$. Note that, for simplicity, we ignore the details of multi-head attention and skip connections between sub-layers, as they do not introduce additional trainable parameters. From a unified view, all sub-layers can be abstracted by a transformation over the sub-layers input,

$$\mathbf{H}_{\text{out}} = m(\mathbf{H}_{\text{in}}) \tag{22}$$

# B Details of Experiment Configurations

## B.1 Details of experiments

We apply $S^3$Delta to the datasets to multitask benchmarks GLUE [38] and SuperGLUE [37] following previous works. All datasets are downloaded from the HuggingFace Datasets [19] library. Since the test split of these datasets are held officially and invisible to the researchers, we randomly split off 2k samples from the training set as validation set $\mathcal{D}_{\text{val}}$ for large datasets(QQP, QNLI, ReCoRD, SST2, MNLI), and use the remaining as the training set $\mathcal{D}_{\text{train}}$, and use the original validation set as the test set $\mathcal{D}_{\text{test}}$. For other datasets, we randomly split the original validation set in half as the validation $\mathcal{D}_{\text{val}}$ and the test set $\mathcal{D}_{\text{test}}$, and use the training set as $\mathcal{D}_{\text{train}}$. The same dataset is splited differently with different random seeds. For each experiment setting, we repeat the experiment with 8 seeds. In all experiments, the maximum sequence length is 128 for the tasks in GLUE and 256 for the tasks in SuperGLUE. The batch size is 16 for SuperGLUE and 32 for GLUE. Especially, we set the maximum sequence length to 512 and batch size to 8 for ReCoRD. We use T5$_{\text{large}}$ model(703M parameters) as the backbone model and we freeze the pre-trained parameters in all experiments except finetuning. We use AdamW as the optimizer with a linear learning rate decay schedule.

For $S^3$Delta, following DARTS [22], we equally split the original training set $\mathcal{D}_{\text{train}}$ into two parts: $\mathcal{D}_{\delta}$ for optimizing the parameters in DT modules, $\mathcal{D}_{\alpha}$ for optimizing the structural parameter. The original validation set, is used to evaluate and save the search structure every $I_{\text{eval}}$ steps. The searched structure is retrained in the original training set $\mathcal{D}_{\text{train}}$, and evaluated in $\mathcal{D}_{\text{val}}$. We report the average performances and standard deviations on the final $\mathcal{D}_{\text{test}}$ across 8 seeds.

## B.2 Hyperparameters

We do pre-experiments on BoolQ and SST2 using learning rates in {3e-5, 3e-4, 3e-3}, $\alpha$ learning rate in {1e-3, 1e-2, 1e-1, 1}, and $\tau$ in {0.1, 0.3, 1} and select the learning rate of 3e-4, alpha learning rate

Table 3: Specific parameters on different tasks.

| | Search | | | Re-train | | |
|---|---|---|---|---|---|---|
| | Batchsize | Epoch | Validation Steps | Batchsize | Epoch | Validation Steps |
| GLUE | | | | | | |
| CoLA | 32 | 15 | 100 | 32 | 15 | 100 |
| MNLI | 32 | 1 | 200 | 32 | 3 | 500 |
| MRPC | 32 | 20 | 50 | 32 | 20 | 50 |
| QNLI | 32 | 4 | 200 | 32 | 4 | 200 |
| QQP | 32 | 1 | 200 | 32 | 3 | 500 |
| SST2 | 32 | 5 | 150 | 32 | 5 | 150 |
| STSB | 32 | 40 | 100 | 32 | 40 | 100 |
| SuperGLUE | | | | | | |
| BoolQ | 16 | 15 | 200 | 16 | 15 | 200 |
| CB | 16 | 60 | 20 | 16 | 60 | 20 |
| COPA | 16 | 40 | 20 | 16 | 40 | 20 |
| MultiRC | 16 | 5 | 200 | 16 | 10 | 200 |
| ReCoRD | 8 | 1 | 200 | 8 | 1 | 200 |
| RTE | 16 | 20 | 50 | 16 | 40 | 50 |
| WiC | 16 | 20 | 100 | 16 | 20 | 100 |

Table 4: The metrics we used to evaluate the GLUE and SuperGLUE Benchmark.

| | Tasks | Metric |
|---|---|---|
| GLUE | CoLA | Matthew's Corr |
| | SST-2 | Accuracy |
| | MRPC | F1 |
| | QQP | F1 |
| | STS-B | Pearson Corr |
| | MNLI | Accuracy |
| | QNLI | Accuracy |
| SuperGLUE | BoolQ | Accuracy |
| | CB | Accuracy |
| | COPA | Accuracy |
| | MultiRC | F1 |
| | ReCoRD | F1 |
| | RTE | Accuracy |
| | WiC | Accuracy |

of 0.1, and $\tau$ of 1 which perform the best. For $\beta$, we set it to 1. The specific parameters on different tasks are listed in Table 3. Specifically, for fine-tuning, we try learning rates in {3e-5, 1e-4, 3e-4} and find that 3e-5 performs the best. We apply these hyperparameters to all baselines and the re-training phase of our searched structure in Table 2 and Figure 2 and conduct no further hyperparameter-tuning. Therefore, the comparison is fair despite that better performance might be achieved with dataset-specific grid search. The metrics we used to evaluate the GLUE and SuperGLUE Benchmark are in Table 4.

## B.3 Computing Resources

We run all the experiments on NVIDIA V100 32GB GPUs.

## C Theoretical Issues about S$^3$Delta

### C.1 Binary Concrete Distribution

We use the Binary Concrete Distribution as a soft approximation to Bernoulli Distribution $z \sim B(1, p)$, where $p$ is the probability of the sample being 1.

$$\tilde{z} = \sigma \left( \frac{1}{\beta} \log \frac{up}{(1-u)(1-p)} \right).$$

(23)

The probability of $\tilde{z} > 0.5$ is $p$, that is when we sample $\tilde{z}$, we can use $\tilde{z}$ to determine $z$'s value,

$$p(\tilde{z} > 0.5) = p \left( \frac{up}{(1-u)(1-p)} > 1 \right) = p(u > 1 - p) = p.$$

(24)

As $\beta$ approaching 0, the Binary Concrete Distribution converges in distribution to the Bernoulli distribution. For any constant $0 < \epsilon < 1$,

$$\lim_{\beta \to 0} P(\tilde{z} < \epsilon) = 1 - p$$

(25)

$$\lim_{\beta \to 0} P(\tilde{z} > 1 - \epsilon) = p,$$

(26)

Therefore $\tilde{z} \xrightarrow{d} z$.

### C.2 Gradient of Global Shifted Sigmoid

In *global shifted sigmoid* function, we multiply a constant with value 1 to the shifted sigmoid to enable a global comparison among the structural parameters of each DT module,

$$p_i = \lambda_i \tilde{p}_i$$

(27)

$$\lambda_i = \frac{\sum_j \text{Detach}(\tilde{p}_j)}{\sum_j \tilde{p}_j} = 1.$$

(28)

Though the value of $p_i$ is equal to the value of $\tilde{p}_i$, The gradient using these two parameterization is different, which result in different behaviour in the optimization of structural parameters.

Using $\tilde{p}_i$ as the parameterization function, the gradient to $\alpha_i$ is

$$\frac{\partial \mathcal{L}}{\partial \alpha_i} = \frac{\partial \tilde{p}_i}{\partial \alpha_i} \frac{\partial \mathcal{L}}{\partial \tilde{p}_i};$$

(29)

while using $p_i$ as the parameterization function, the gradient is

$$\frac{\partial \mathcal{L}}{\partial \alpha_i} = \sum_j \frac{\partial \mathcal{L}}{\partial p_j} \frac{\partial p_j}{\alpha_i} = \sum_{j \neq i} \frac{\partial \mathcal{L}}{\partial p_j} \frac{\partial p_j}{\partial \alpha_i} + \frac{\partial \mathcal{L}}{\partial p_i} \frac{\partial p_i}{\partial \alpha_i}$$

(30)

$$= \sum_j \frac{\partial \mathcal{L}}{\partial p_j} \frac{\partial \lambda_j}{\partial \alpha_i} \tilde{p}_j + \lambda_j \frac{\partial \mathcal{L}}{\partial p_i} \frac{\partial \tilde{p}_i}{\partial \alpha_i}$$

(31)

$$= \sum_j \left( \frac{-\sum_k \text{Detach}(\tilde{p}_k)}{(\sum_k \tilde{p}_k)^2} \frac{\partial \tilde{p}_i}{\partial \alpha_i} \tilde{p}_j \frac{\partial \mathcal{L}}{\partial p_j} \right) + \frac{\partial \mathcal{L}}{\partial p_i} \frac{\partial \tilde{p}_i}{\partial \alpha_i}$$

(32)

$$= \frac{\partial \tilde{p}_i}{\partial \alpha_i} \left( -\sum_j \frac{\tilde{p}_j}{\sum_k \tilde{p}_k} \frac{\partial \mathcal{L}}{\partial p_j} + \frac{\partial \mathcal{L}}{\partial p_i} \right).$$

(33)

The additional term

$$-\sum_j \frac{\tilde{p}_j}{\sum_k \tilde{p}_k} \frac{\partial \mathcal{L}}{\partial p_j}$$

(34)

serves as an adjustment from the global structural gradient to the local gradient. In a special case, if the gradients to all $p_i$, i.e., $\frac{\partial \mathcal{L}}{\partial p_i}$, are equal, no gradient will be pass to the structural parameter $\alpha$, which is reasonable.

We conduct ablation studies on three datasets using the global shifted sigmoid $p_i$ (denoted by Global) and the shifted sigmoid without global comparison $\tilde{p}_i$ (denoted by Local). Figure 8 proves the correctness of the global shifted sigmoid parameterization.

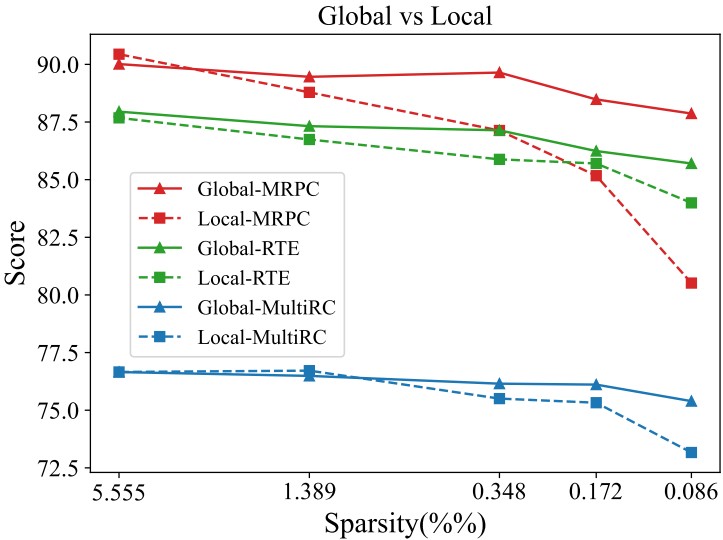

Figure 8: The performance difference between local and global shifted sigmoid.

Table 5: The computational resources in the searching phase and re-training phase, Computation time, memory consumption, and the corresponding ratio between searching phase and re-training phase are listed.

| Dataset | Time/min | | | Memory/GB | | |
|---------|--------|----------|-------|--------|----------|-------|
|         | Search | Re-train | Ratio | Search | Re-train | Ratio |
| RTE     | 148.6  | 30.0     | 5.0   | 27.7   | 16.6     | 1.7   |
| STSB    | 139.3  | 26.3     | 5.3   | 28.9   | 10.6     | 2.7   |
| CoLA    | 145.6  | 17.0     | 8.6   | 16.1   | 8.9      | 1.8   |

# D    Additional Results of Experiments

## D.1    Efficiency of S$^3$Delta

We report the detailed consumption of computational resources in S$^3$Delta . From Table 5, we can see that the searching time is approximately 5∼9 times of a training time, which is acceptable due to the large search space. Once a sparse structure is searched, the training and inference are as fast as or faster than the other DT methods.

## D.2    Heat maps of $p_i$ on Each Datasets

The heat maps of the $p_i$ on different datasets are presented in Figure 9. Most of the datasets follow the similar trend in the activated positions. STSB and ReCoRD's optimal solution is a bit different from the others, which can be further investigated.

# E    Potential societal Impact

S$^3$Delta focuses on developing a method to find the optimal sparse structure of DT modules. The searching process takes longer time than a single training process, consuming more energy. However, the searched structures are found to be highly transferable. Therefore, we can search for an approximately optimal solution on a group of similar tasks and reuse the structure on other unseen tasks. Consequently, S$^3$Delta is environmental friendly. Another potential negative societal impact is the malicious injection of DT modules, which will potentially harm the adaption performance of PTMs.

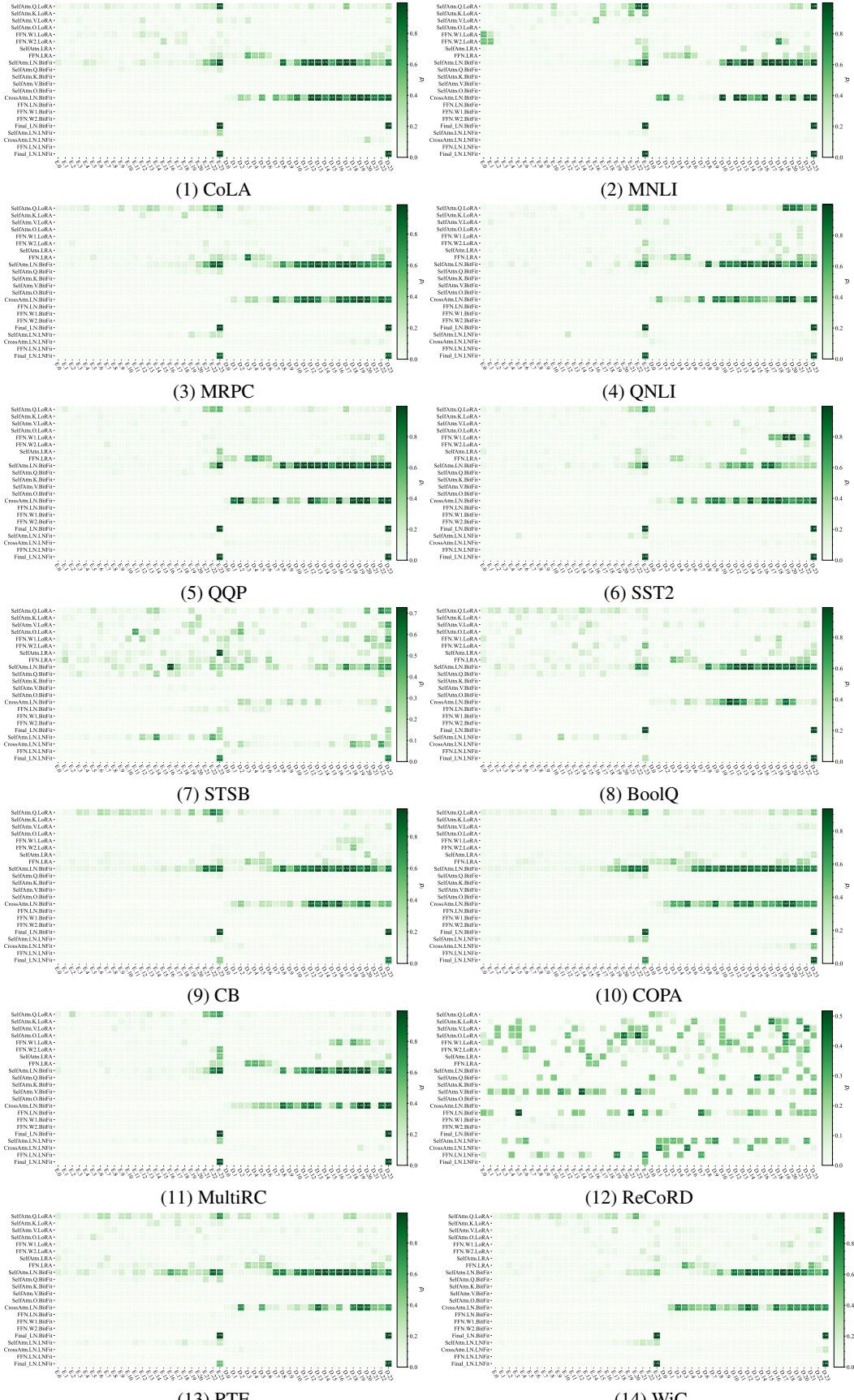

Figure 9: Heat maps of $p_i$ on each datasets