# OpenReview forum: "Sparse Structure Search for Delta Tuning"
_NeurIPS.cc/2022/Conference — NeurIPS 2022 Accept_

### Official Review · Reviewer_qZC4 · 2022-07-10

**Rating:** 6
**Confidence:** 3
**Soundness:** 3 good
**Presentation:** 3 good
**Contribution:** 3 good

**Summary:**

The paper presents S3PET an approach to automatically search optimal trainable parameter efficient structure within language models. The proposed approach consists of three main components: A unified search space with probabilistic gating, a differentiable parameter efficient training search algorithm and an explicit sparsity control algorithm. Results on GLUE and SuperGLUE benchmarks show competitive performance with SOTA while using less number of trainable parameters.

**Questions:**

The authors do not compare with Prompt Tuning baselines and they mention that it takes much longer to converge [L208].
If there are some initial experiments about this, it would be good to report numbers on how long it would actually take to converge.

**Limitations:**

The authors did not provide any explicit discussions on limitations. For example in terms of efficiency of training, the approach seems to be much slower than pre-defined sparse structures.

**Strengths And Weaknesses:**

Strengths:
- Unlike predefined trainable substructures in previous works, the proposed approach searches the optimal trainable structure and is guided by downstream performance.
- One important advantage of the approach seems to be the strong results even when the number of training parameters is small (~0.01%).
This is significantly less than prior methods.

Weaknesses:
- The experiments only focus on GLUE/SuperGLUE classification tasks. There are no results on generation tasks like translation or summarization.
- It would have been interesting to see some analysis on training data requirements for the proposed approach. E.g., would it still work if there are limited data.
- The only pretrained model experimented with is T5-large. Performance under larger models or decoder-only models is unexplored.
- It would have been interesting to see some discussion on how expensive the training of the proposed approach is. There is some discussion in the appendix but I would encourage authors to move those to main text.

---

> ### Author Response · Authors · 2022-08-02
> **Answer to Missing Tasks, Data Requirements, Few PLMs, Training Overhead, and Prior Experiments.**
>
> Thanks for your recognition of our work!
> 1. **Missing results on generation tasks**. Following the previous PET works like compacter and adapter, we mainly conduct experiments on GLUE and SuperGLUE. We will add the result on generation in our future version.
> 2. **Data requirements**. The SuperGLUE-CB dataset is composed of 250 training samples, 56 validation samples, and 250 test samples, which can be seen as a limited data scenario. And S$^3$PET still achieves the best compared to the methods in the Seach Space. We did observe, however, that a substantial increase in the proportion of tunable parameters can improve the performance of SuperGLUE-CB.
> 3. **Expecting More PLM Types**. Theoretically, our method is independent of the architecture or type of PLMs. We will add experiments in more PLMs in the future versions.
> 4. **Training Overhead**. It is a common question, please refer to the [response to AC and Common Questions](https://openreview.net/forum?id=oOte_397Q4P&noteId=zFIZSTJeWNm)
> 5. **Question: Prompt Tuning prior experiments**. We list the prior experiments of prompt tuning in the table below. The convergence step is defined as the step that achieves the highest performance on the dev set. we will add these tables to the future version of our work.
>
> |                   | **CoLA** | **SST-2** | **MRPC** | **QQP** | **STSB** | **MNLI** | **QNLI** |
> |-------------------|----------|-----------|----------|---------|----------|----------|----------|
> | Convergence Steps | 20900    | 23100     | 4950     | 22850   | 17950    | 33250    | 27550    |

---

### Official Review · Reviewer_mV46 · 2022-07-11

**Rating:** 6
**Confidence:** 2
**Soundness:** 2 fair
**Presentation:** 2 fair
**Contribution:** 3 good

**Summary:**

This paper proposes to design an automatic scheme to search the PET models for parameter-efficient finetuning.  Extensive experiments are given, which demonstrates that the auto-searched method surpasses the manual and random ones with less trainable parameters.


**Questions:**

See weakness

**Limitations:**

yes

**Strengths And Weaknesses:**

Strength
---The ideal of this work is interesting, while combines the architecture search and PET models tuning.
The given experiments compared with seveal pior arts, which show the superior of the this proposes method.

Weakness
-- Although this work claims that the trainable parameters are reduced obviously, the runtime of the finetuning should be reported to further verify the effectness of this work.

---

> ### Author Response · Authors · 2022-08-02
> **Answer to Runtime of Fine-tuning.**
>
> Thanks for your positive feedback on our paper!
> 1. **Runtime of fine-tuning.** It is a common question. Please refer to the [response to AC and Common Questions](https://openreview.net/forum?id=oOte_397Q4P&noteId=zFIZSTJeWNm).

---

### Official Review · Reviewer_LCyP · 2022-07-11

**Rating:** 6
**Confidence:** 4
**Soundness:** 3 good
**Presentation:** 3 good
**Contribution:** 3 good

**Summary:**

This paper develops a neural architecture search method for the adapter module (or called PET modules) in parameter-efficient tuning schemes.  The authors construct a unified search space for parameter-efficient tuning, and adopted binary differentiable NAS method for architecture search. A shifted global sigmoid is developed to control the sparsity of the PET structure. The searched PET structure achieves competitive results with about 1/5 trainable parameters compared to previous manually designed methods.

**Questions:**

1. How does PET methods save model adaptation cost? Optimizing an PET module with gradient decent requires all the subsequent network layers to calculate their gradient in the backward pass. All the parameters (after the first PET module) still need to compute and save gradient. Thus, the computational cost and memory cost is actually not reduced.


2. Minor suggestions:

   (1) It would be better to use \eqref in latex for reference of equations (add parentheses for equation numbers).

   (2) Avoid using "Moreover" constantly in consecutive sentences (in Line 66-68). Some words like "Firstly... Secondly..." might be better.

**Limitations:**

Yes.

**Strengths And Weaknesses:**

Strengths:
1. This paper is generally in good form and easy to follow. Figures and tables are professional.

2. The studied problem, automatic design of PET structure, is novel and interesting.

3. The unified search space may be useful for future research; the proposed Shifted Global Sigmoid for sparsity control is sound and effective.

4. The experimental evaluation is extensive, covering transferability of the searched structure, compare of search space, compare of sparsity method, etc.

Weakness:
1.  My main concern is the contradiction between the search cost and the application scenario of S3PET. The motivation of PET method is to reduce the model adaptation cost. However, performing the proposed PET search would cost 5 to 8 times training time (as in line 263), which may contradict the motivation of PET methods. I understand the authors claim of the search cost is comparable to (or cheaper than) manual design. However, similar effort is needed for the proposed method, e.g. on search space design or hyper-parameter tuning. Overall, this issue of search cost could severely limit the application scenarios and the significance of S3PET.

2. Some important literatures are not introduced. (1) The proposed method is highly relevant to differentiable NAS methods with discrete binary gate, e.g. ProxylessNAS [*1], FBNet [*2], etc. In DARTS, the options of gate are activated together in each step, while in these methods, only one option is activated and the structure parameters are optimized by replacing the hard option with soft approximation in backward calculation, same to this paper. (2) Some automatic model adaptation method should be introduced: [*3] [*4].


[*1] Han Cai, et al. ProxylessNAS: Direct Neural Architecture Search On Target Task And Hardware. ICLR, 2019.

[*2] Bichen Wu, et al. FBNet: Hardware-Aware Efficient ConvNet Design via Differentiable Neural Architecture Search. CVPR, 2019.

[*3] Guo, Y., et al. Spottune: transfer learning through adaptive fine-tuning. CVPR, 2019.

[*4] Guo, Y., et al. Adafilter: Adaptive filter fine-tuning for deep transfer learning. AAAI, 2020.

---

> ### Author Response · Authors · 2022-08-02
> **Answer to Search Cost, Missing Literature, Adaptation Cost, and Minor Suggestions**
>
> Thanks for your appreciation of our work!
>
> 1. **Search Cost.** This is a common question. Please refer to the [response to AC and Common Questions](https://openreview.net/forum?id=oOte_397Q4P&noteId=zFIZSTJeWNm). In addition to this common answer, you are concerned that NAS requires a similar amount of effort compared to manual design. However, we believe that search space design is much easier than manually selecting a structure. In fact, we did not spend much effort on the search space design: we just added the legal positions of the corresponding PET methods to the search space. As for the tuning of hyperparameters, we spent some time in the initial design of the whole S3PET algorithm. And after the success of the algorithm, we kept the hyperparameters (e.g., the learning rate of the structural parameters) fixed in all experiments (no tuning for the dataset). We believe that these hyperparameters can be reused to a large extent for future applications.
>
> 2.  **Missing Literature.** We thank you for your extensive knowledge in this field and we add these papers in our revised version of the paper （Line 76 and Line 105）. Your expertise in this area gives great weight to your positive attitude towards our papers. In fact, we have learned a lot from these papers in the design of our algorithms. We believe that the contribution of our algorithm, based on these excellent papers, is the budget control method that explicitly makes the binary gates activated at the desired sparsity level while keeping the training successful. The explicit sparsity control method makes our binary gate enjoy advantages similar to Softmax gates.
>
> 3. **Adaptation Cost.** The adaptation cost comes from two aspects: saving storage, and training efficiency. Since only a small fraction of the parameters are updated, we can save only the updated parameters. As for the training efficiency, the local gradient information is stored in the optimizer and used to compute the parameter updates. If we use an optimizer like Adam, the runtime GPU memory is about 4 times [*1] more than the memory needed to hold the model parameters (even at batchsize=1). With PET, the GPU memory is reduced to a little more than 1x the model parameters when using small batches [*2]. This is because the number of parameters to be updated is very small, thus the local gradients of the majority of parameters don't need to be stored in memory.
>
> 4. **Minor Suggestions.** We really appreciate the effort and care you put into reading our papers. We have modified those small suggestions accordingly.
>
> [*1] Rajbhandari, S., et al. Zero: Memory optimizations toward training trillion parameter models. SC20.
>
> [*2] Ding, Ning, et al. Delta tuning: A comprehensive study of parameter efficient methods for pre-trained language models. Preprint 2022.

---

### Official Review · Reviewer_SumM · 2022-07-12

**Rating:** 5
**Confidence:** 4
**Soundness:** 3 good
**Presentation:** 2 fair
**Contribution:** 2 fair

**Summary:**

This paper explores constructing PET modules in an automatic manner, and S 3PET with NAS is proposed, which conducts the differentiable PET structure search through bi-level optimization and proposes shifted global sigmoid method to explicitly control the number of trainable parameters. Extensive experiments show superior performance with low trainable parameters budgets.

**Questions:**

As listed in Weaknesses, the following questions need to be solved.

(1)	Evaluation details and oracle.

(2)	Experiments details and more explanations.


**Limitations:**

There is not serious negative societal impact of this work currently.

**Strengths And Weaknesses:**

Pros:

(1)	Automated search for parameter-efficient tuning.

(2)	State-of-the-art performance with low trainable parameters.

Cons:

Although the authors employ NAS for parameter-efficient tuning, some descriptions are not clear.

(1)	The authors state that they choose the set of positions where the sum of p_i. However, how to determine the number of positions? If all positions are used, does the performance improve? It shows the oracle.

(2)	In Table1, the authors show the results on GLUE and SuperGLUE. However, which dataset is employed for NAS? Is it searched on GLUE and SuperGLUE separately? If so, I still think it lacks generalization ability. Adding NAS step is also time-consuming, the authors should also report the time cost of NAS compared to the training step. In Table2, it seems that the performance improvement is not significant. It is obvious that NAS can improve performance, so it lacks insight. In Table 3, the authors conduct the transferability experiments, but it is not discussed why such a method can guarantee transferability. Maybe it is because the dataset for searching must be very large.

---

> ### Author Response · Authors · 2022-08-02
> **Answer to Positions, Oracle, Generalization, and Cost.**
>
> Thank you for your insightful questions! Here we answer them in detail.
> 1. **The number of positions that we should choose.** We first elaborate on how we choose the final activated positions more clearly. We select the top modules with the highest probability and ensure that the number of parameters in the selected modules just reaches the budget, as mentioned in the paper. And in terms of how to determine the budget. Our answer is that the budget depends on the user's needs. For example, imagine a situation where a user wants to train thousands of PET modules in a personalized dialog setting, wants to learn PET modules for each long document or wants to choose from many PET checkpoints. They can calculate the storage quota that each PET checkpoint can occupy to determine the budget of trainable parameters and the number of locations to use.
> 2. **Oracle performance**. If all locations are used, we verified that this leads to performance saturation, i.e., the performance degradation caused by using only a small fraction of locations is negligible. Therefore, we tend to search for the most efficient modules. We provide the performance of Oracle in the following table.
>
>
> |          | **Parameter Ratios** | **CoLA**       | **SST-2**      | **MRPC**       | **QQP**        | **STSB**       | **MNLI**       | **QNLI**       | **AVG** |
> |----------|----------------------|----------------|----------------|----------------|----------------|----------------|----------------|----------------|---------|
> | Oracle   | 25.51%%              | 59.35$\pm$5.74 | 95.81$\pm$0.60 | 92.71$\pm$0.50 | 88.54$\pm$0.13 | 91.51$\pm$0.39 | 89.56$\pm$0.24 | 93.93$\pm$0.32 | 87.34   |
> | S$^3$PET | 1.39%%               | 59.34$\pm$4.75 | 95.84$\pm$0.14 | 92.13$\pm$2.09 | 88.04$\pm$0.23 | 91.58$\pm$0.25 | 89.14$\pm$0.13 | 94.12$\pm$0.12 | 87.17   |
>
> 3. **Generalization.** The focus on generalization is indeed very valuable. Before we illustrate the generalization issues, we clarify our experiment settings a bit. In the experiments in Table 2, we searched with the same dataset as in the test but using the validation set. We neither searched on a mixture of several datasets from the GLUE benchmark or the SuperGLUE benchmark in Table 2 nor did we perform transfer learning. This is because Table 2 is intended to demonstrate that our method can improve parameter efficiency in the i.i.d. setting. To demonstrate generalizability or transferability, we conducted transferable experiments in Table 3 and obtained positive results. The conclusion of the transferable experiments in Table 3 is that we can obtain *non-trivial* results after transfer. Although this does not mean that the structure after the transfer is optimal, because optimality after the transfer is not guaranteed theoretically. For the reason of non-trivial transferability, we believe that it comes from the *similarity of the source and target tasks* since they are both tasks that require semantic knowledge. Heat maps of $p_i$ on each dataset (See Appendix Figure 8)) also show the similarity of the different structures, which guarantees transferability. Further experiments investigating the dependence of optimal structure and task type deserve to be studied in detail in the future (beyond the scope of this work), which could even serve as a probe into the functionality of sub-modules of PLMs.
> 4. **Cost of NAS steps**. It's a common question, please refer to [answers to Common Questions](https://openreview.net/forum?id=oOte_397Q4P&noteId=zFIZSTJeWNm)

---

> > ### Comment · Reviewer_SumM · 2022-08-09
> > **Response to Authors**
> >
> > Thanks for the authors’ response.
> >
> > As shown in my initial review, I still think that the performance improvement is not significant. It is obvious that NAS can improve performance, so it lacks insight. However, the authors have solved my part concerns (e.g., oracle, cost of NAS). I would update my rating from 4 to 5.

---

### Author Response · Authors · 2022-08-02
**To Area Chair and Answers to Common Questions**


We greatly appreciate the efforts of AC in organizing the review of our paper. We think all the reviewers are very professional and treat their questions and suggestions very seriously. We have responded thoroughly to each and every point of the reviewer's comments. Here we answer some common questions in a unified way.
1. **Training efficiency of NAS**. The times for the NAS steps were originally listed in Appendix D.1 and the analysis was in Section 4.7. And we *have moved them to the body of the paper in the revised version* (See Table 3 and Section 4.7 in the updated rebuttal revision). In general, running S3PET requires 5 to 8 times the training time, which we agree is not a very efficient time overhead. However, we do believe that our approach still has strong merits based on the following points.

- First, finding an optimal structure and training it will naturally take more time than training a predefined structure. This is because if we allow a structure to be optimizable, the *optimization space increases significantly*. And this larger optimization space is where the performance improvement comes from.
- Second, we are, in fact, the first to *verify that NAS runtimes are acceptable in large pre-trained models*. Our work encourages the development of specialized and more efficient NAS algorithms conditioned on large PLMs.
- Third, the searched structures are *transferable*. Therefore, when the structures are reused, no additional training overhead will be introduced compared to the pre-defined structures.
- Last but not least, the efficiency of the PET approach comes from two aspects: storage efficiency, and training efficiency. We significantly *reduce the storage efficiency* and thus potentially favor the case of training many different PET modules.

---

### Meta-Review · Area_Chair_pGoh · 2022-08-23

**Recommendation:** Accept
**Confidence:** Certain

**Metareview:**

This paper presents work on parameter-efficient tuning of large pre-trained model.  The main contribution is an automated search for the parameter efficient tuning modules, in a neural architecture search style.

The reviewers raised questions regarding the overall efficiency of the training scheme given the cost to the neural architecture search.  However, they believed that the empirical results and overall novelty of the approach together were a solid contribution to research in this area.  The additional clarifications brought into the main text help to make these efficiency concerns clearer and better position the work.  Based the overall novelty of approach and results, this paper is ready for publication in NeurIPS.

**Award:**

No

---

### Decision · Program_Chairs · 2022-09-14

Accept